# ERMAS: Learning Policies Robust to Reality Gaps in Multi-Agent Simulations

## Abstract

Policies for real-world multi-agent problems, such as optimal taxation, can be learned in multi-agent simulations with AI agents that emulate humans. However, simulations can suffer from reality gaps as humans often act suboptimally or optimize for different objectives (i.e., bounded rationality). We introduce $\epsilon$-Robust Multi-Agent Simulation (ERMAS), a robust optimization framework to learn AI policies that are robust to such multi-agent reality gaps. The objective of ERMAS theoretically guarantees robustness to the $\epsilon$-Nash equilibria of other agents – that is, robustness to behavioral deviations with a regret of at most $\epsilon$. ERMAS efficiently solves a first-order approximation of the robustness objective using meta-learning methods. We show that ERMAS yields robust policies for repeated bimatrix games and optimal adaptive taxation in economic simulations, even when baseline notions of robustness are uninformative or intractable. In particular, we show ERMAS can learn tax policies that are robust to changes in agent risk aversion, improving policy objectives (social welfare) by up to 15% in complex spatiotemporal simulations using the AI Economist (Zheng et al., 2020).

## 1 Introduction

Reinforcement learning (RL) offers a tool to optimize policy decisions affecting complex, multi-agent systems; for example, to improve traffic flow or economic productivity. In practice, the need for efficient policy evaluation necessitates training on simulations of multi-agent systems (MAS). Agents in these systems can be emulated with fixed behavioral rules, or by optimizing for a reward function using RL (Zheng et al., 2020). For instance, the impact of economic policy decisions are often estimated with agent-based models (Holland & Miller, 1991; Bonabeau, 2002). This commonly introduces a *reality gap* as the reward function and resulting behavior of simulated agents might differ from those of real people (Simon & Schaeffer, 1990). This becomes especially problematic as the complexity of the simulation grows, for example, when increasing the number of agents, or adding agent affordances (Kirman, 1992; Howitt, 2012). As a result, policies learned in imperfect simulations need to be robust against reality gaps in order to be effective in the real world.

We introduce $\epsilon$-Robust Multi-Agent Simulation (ERMAS), a robust optimization framework for training robust policies, termed *planners*, that interact with real-world multi-agent systems. ERMAS trains robust planners by simulating multi-agent systems with RL and sampling *worst-case* behaviors from the *worst-case* agents. This form of multi-agent robustness poses a very challenging multi-level (e.g., max-min-min) optimization problem. Existing techniques which could be applied to ERMAS's multi-agent robustness objective, e.g., naive adversarial robustness (Pinto et al., 2017) and domain randomization (Tobin et al., 2017; Peng et al., 2018), are intractable as they would require an expensive search through a large space of agent reward functions. Alternative frameworks improve robustness, e.g., to changes in environment dynamics, observation or action spaces (Pinto et al., 2017; Li et al., 2019; Tessler et al., 2019), but do not address reality gaps due to reward function mismatches, as they use inappropriate metrics on the space of adversarial perturbations.

To solve this problem, ERMAS has three key features: 1) It formulates a multi-agent robustness objective equivalent to finding the worst case $\epsilon$-Nash equilibria. 2) It optimizes a tractable dual problem to the equivalent objective. 3) It approximates the dual problem using local solution concepts and first-order meta-learning techniques (Nichol et al., 2018; Finn et al., 2017). ERMAS ultimately yields policies that are robust to other agents' behavioral deviations, up to a regret of $\epsilon$.

We show that ERMAS learns robust policies in repeated bimatrix games by finding the worst-case reality gaps, corresponding to highly adversarial agents, which in turn leads to more robust planners. We further consider a challenging, large-scale spatiotemporal economy that features a social planner that learns to adjust agent rewards. In both settings, we show policies trained by ERMAS are more robust by testing them in perturbed environments with agents that have optimized for reward functions unused during ERMAS training. This generalization error emulates the challenge faced in transferring policies to the real world. In particular, we show ERMAS can find AI Economist tax policies that achieve higher social welfare across a broad range of agent risk aversion objectives. In all, we demonstrate ERMAS is effective even in settings where baselines fail or become intractable.

**Contributions** To summarize, our contributions are:

- We derive a multi-agent adversarial robustness problem using $\epsilon$-Nash equilibria, which poses a challenging nested optimization problem.

- We describe how ERMAS efficiently solves the nested problem using dualization, trust-regions, and first-order meta-learning techniques.

- We empirically validate ERMAS by training robust policies in two multi-agent problems: sequential bimatrix games and economic simulations. In particular, ERMAS scales to complex spatiotemporal multi-agent simulations.

## 2 ROBUSTNESS AND REALITY GAPS IN MULTI-AGENT ENVIRONMENTS

We seek to learn a policy $\pi_p$ for an agent, termed the *planner*, that interacts with an environment featuring $N$ other agents. The planner's objective depends both on its own policy and the behavior of other agents in response to that policy; this is a multi-agent RL problem in which the planner and agents co-adapt. In practice, evaluating (and optimizing) $\pi_p$ requires use of a simulation with agents that emulate those in the environment of interest (i.e. the real world), which might contain agents whose reward function differs from those used in the simulation. Our goal is to train planner policies that are robust to such reality gaps.

Formally, we build on partially-observable multi-agent Markov Games (MGs) (Sutton & Barto, 2018), defined by the tuple $M := (S, A, r, \mathcal{T}, \gamma, o, \mathcal{I})$, where $S$ and $A$ are the state and action spaces, respectively, and $\mathcal{I}$ are agent indices. Since the MG played by the agents depends on the choice of planner policy, we denote the MG given by $\pi_p$ as $M[\pi_p]$. MGs proceed in episodes that last $H + 1$ steps (possibly infinite), covering $H$ transitions. At each time $t \in [0, H]$, the world state is denoted $s_t$. Each agent $i = 1, \ldots, N$ receives an observation $o_{i,t}$, executes an action $a_{i,t}$ and receives a reward $r_{i,t}$. The environment transitions to the next state $s_{t+1}$, according to the transition distribution $\mathcal{T}(s_{t+1}|s_t, \boldsymbol{a}_t)$.[1] Each agent observes $o_{i,t}$, a part of the state $s_t$. Agent policies $\pi_i$ are parameterized by $\theta_i$ while the planner policy $\pi_p$ is parameterized by $\theta_p$.

The Nash equilibria of $M[\pi_p]$ are agent policies where any unilateral deviation is suboptimal:

$$\text{ANE}(\pi_p) := \{\boldsymbol{\pi} \mid \forall i \in [1, N], \tilde{\pi}_i \in \Pi : J_i(\tilde{\pi}_i, \pi_{-i}, \pi_p) \leq J_i(\pi_i, \pi_{-i}, \pi_p)\}, \quad (1)$$

where $J_i(\boldsymbol{\pi}, \pi_p) := \mathbb{E}_{\boldsymbol{\pi}, \pi_p}\left[\sum_{t=0}^{H} \gamma^t r_t^{(i)}\right]$ denotes the objective of agent $i$. Hence, a rational agent would not unilaterally deviate from $\boldsymbol{\pi} \in \text{ANE}(\pi_p)$.

To evaluate a fixed planner policy $\pi_p$, we simply sample outcomes using policies $\boldsymbol{\pi} \in \text{ANE}(\pi_p)$. Also optimizing $\pi_p$ introduces a form of *two-level learning*. Under appropriate conditions, this can be solved with simultaneous gradient descent (Zheng et al., 2020; Fiez et al., 2019).

**Robustness Objective** As noted before, we wish to learn planner policies $\pi_p$ that are robust to reality gaps arising from changes in agent reward functions, e.g., when agents are boundedly rational.[2] We develop a robustness objective for the planner by formalizing such reality gaps as perturbations

---

[1] Bold-faced quantities denote vectors or sets, e.g., $\boldsymbol{a} = (a_1, \ldots, a_N)$, the action profile for $N$ agents.

[2] This type of reality gap occurs when the simulated environment's reward function $r$ fails to rationalize the actual behavior of the agents in the real environment, i.e., when agents in the real world act suboptimally with respect to the simulation's reward function.

$\xi_i \in \Xi$ to agent objectives, where the *uncertainty set* $\Xi : (S, A)^H \to \mathbb{R}$ is the space of possible perturbations and represents uncertainty about the objectives of other agents. We extend $\text{ANE}(\pi_p, \boldsymbol{\xi})$ to condition the Nash equilibria on perturbations $\boldsymbol{\xi}$:

$$\text{ANE}(\pi_p, \boldsymbol{\xi}) := \{ \boldsymbol{\pi} \mid \forall i \in [1, N], \tilde{\pi}_i \in \Pi : J_i^\xi(\tilde{\pi}_i, \pi_{-i}, \pi_p) \leq J_i^\xi(\pi_i, \pi_{-i}, \pi_p) \}, \quad (2)$$

$$J_i^\xi(\tilde{\pi}_i, \pi_{-i}, \pi_p) := J_i(\tilde{\pi}_i, \pi_{-i}, \pi_p) + \mathbb{E}_{\tau_i \sim \tilde{\pi}_i, \pi_{-i}, \pi_p} [\xi_i(\tau_i)] \quad (3)$$

where $\tau_i$ is a trajectory (sequence of state-action pairs). Following Morimoto & Doya (2001), a robust planner optimizes its reward, subject to agents playing a perturbed Nash equilibrium $\text{ANE}(\pi_p, \boldsymbol{\xi})$ that maximally penalizes the planner:

$$\pi_p^* = \arg\max_{\pi_p} \min_{\boldsymbol{\xi} \in \Xi} \min_{\boldsymbol{\pi} \in \text{ANE}(\pi_p, \boldsymbol{\xi})} J_p(\boldsymbol{\pi}, \pi_p). \quad (4)$$

Note that agent policies $\boldsymbol{\pi} \in \text{ANE}(\pi_p, \boldsymbol{\xi})$ describes agents that optimize their own reward function, and we assume an adversary chooses $\boldsymbol{\xi}$.

**Bounded Uncertainty Set**   There are two challenges with Equation 4. First, if the adversary can arbitrarily choose $\Xi$, the worst case is uninformative.[3]  Second, depending on the complexity of $\Pi$, the uncertainty set $\Xi$ may be high-dimensional and intractable to search. We address these issues by upper-bounding the size of the uncertainty set, $L_\infty$ norm of $\xi_i \in \Xi$, by the term $\epsilon$. Thus $\epsilon$ upper-bounds the difference between the reward functions of agents in the training and testing environments, e.g., between simulation and the real world. This bounded uncertainty set is:

$$\Xi_\epsilon := \left\{ \xi \ \middle| \ \sup_{\boldsymbol{\pi}, \pi_p} |\xi_i(\boldsymbol{\pi}, \pi_p)| < \epsilon, \text{ for all } i \in I \right\}. \quad (5)$$

This uncertainty set is equivalent to the $\epsilon$-equilibria of $M[\pi_p]$:

$$\text{ANE}(\pi_p, \epsilon) := \{ \boldsymbol{\pi} \mid \forall i \in [1, N], \tilde{\pi}_i \in \Pi : J_i(\tilde{\pi}_i, \pi_{-i}, \pi_p) \leq J_i(\pi_i, \pi_{-i}, \pi_p) + \epsilon \}. \quad (6)$$

$\epsilon$ is a tunable hyperparameter—this is the case with most robust RL (Pinto et al., 2017; Li et al., 2019)—but a good starting value is the anticipated error in reward objective estimates (application-specific). Using 6, the robustness objective becomes the following constrained optimization problem:

$$\underbrace{\arg\max_{\pi_p} J_{p,\min}^*(\pi_p, \epsilon)}_{\texttt{Planner-OPT}}, \quad \text{where } J_{p,\min}^*(\pi_p, \epsilon) := \underbrace{\min_{\boldsymbol{\pi} \in \text{ANE}(\pi_p, \epsilon)} J_p(\boldsymbol{\pi}, \pi_p)}_{\texttt{Agent-Adv-Search}}. \quad (7)$$

Using $\text{ANE}(\pi_p, \epsilon)$ replaces the problem of intractably searching through $\Xi$ with searching through $\text{ANE}(\pi_p, \epsilon)$, and thus merges the two nested $\min$ operations in Equation 4. Conceptually, this transfers the worst-case search problem to the agents: `Agent-Adv-Search` agents find an adversarial equilibrium; `Planner-OPT` optimizes the planner given adversarial agents. Note that the constraint set in Eq 7 is non-empty for $\epsilon \geq 0$; the constraints (Eq 6) simply upper-bound the regret of agents. By definition, for non-empty bounded $\Pi$, there exists an optimal policy with zero regret.

## 3   ERMAS: ROBUST POLICIES IN MULTI-AGENT SIMULATIONS

We now introduce ERMAS, an efficient optimization framework to solve the robustness problem in Equation 7. ERMAS proceeds in three steps. First, it dualizes Equation 7 following constrained RL. Second, it defines a trust-region for the uncertainty set $\text{ANE}(\pi_p, \epsilon)$, approximating the dual problem. Finally, it uses first-order meta-learning with trust-regions to solve the approximate dual problem. **See Appendix A.1 for the detailed Algorithm description.**

**Dualizing `Agent-Adv-Search`**   The agent search problem in Equation 7 can be formulated similar to a constrained RL problem (Paternain et al., 2019), where the primary objective of the agents is to minimize the planner's reward and the secondary objective is to maximize their own reward. While conventional constrained RL enforces a constant lower bound in the constraint, e.g.,

---

[3]For instance, by setting $\xi_i$ such that $J_i^\xi = -J_p$.

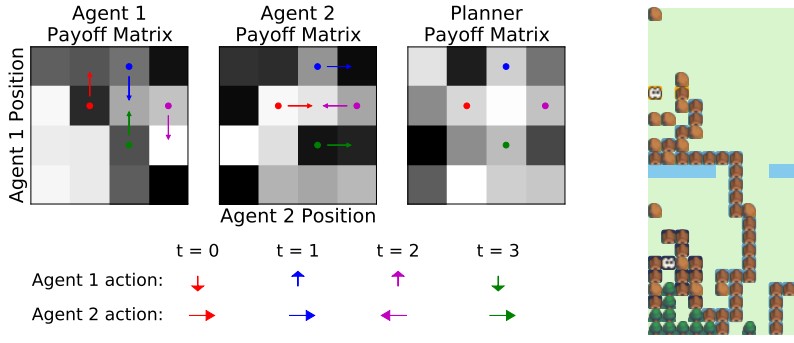

(a) Repeated Bimatrix Game  (b) Spatiotemporal Economic Simulation

Figure 1: ERMAS trains planners that are robust to reality gaps in these testbed multi-agent simulations. **(Left)** In the repeated bimatrix game, a pair of agents navigates a 2D landscape. Both agents and the planner receive rewards based on visited coordinates. Brighter squares indicate higher payoff. **(Right)** In the spatiotemporal economic simulation, 4 heterogenous agents perform labor, trade resources, earn income, and pay taxes according to the schedules set by the planner.

$J_i(\boldsymbol{\pi}, \pi_p) \geq C$, we enforce a dynamic one: $\forall i \in 1 \ldots N : J_i(\boldsymbol{\pi}, \pi_p) \geq J_i(\pi_i^*, \pi_{-i}, \pi_p) - \epsilon$, where $\pi_i^*$ is the optimal unilateral deviation for agent $i$: $\pi_i^* := \arg\max_{\tilde{\pi}_i \in \Pi} J_i(\tilde{\pi}_i, \pi_{-i}, \pi_p)$.

Letting $\boldsymbol{\lambda}$ denote Lagrange multipliers, we can dualize `Agent-Adv-Search`, i.e., Equation 4, as:

$$\min_{\boldsymbol{\pi}} \underbrace{\left( J_p(\boldsymbol{\pi}, \pi_p) - \sum_{i=1}^{N} \lambda_i \left[ J_i(\pi_i^*, \pi_{-i}, \pi_p) - J_i(\boldsymbol{\pi}, \pi_p) - \epsilon_i \right] \right)}_{J_{p,\min}^{\dagger}(\pi_p, \epsilon)}, \quad (8)$$

This is identical to the dualization of constrained reinforcement learning, whose duality gap is empirically negligible and provably zero under weak assumptions (Paternain et al., 2019). We now abuse notation to denote $\boldsymbol{\theta} := [\theta_1, \ldots, \theta_N, \theta_p]$ and $J_p(\boldsymbol{\theta}) := J_p(\boldsymbol{\pi}, \pi_p)$ where $\theta$ are the parameters of $\pi$. To solve Equation 8, the agents apply gradients:

$$\nabla_{\theta_i} J_{p,\min}^{\dagger}(\pi_p, \epsilon) = -\nabla_{\theta_i} J_p(\boldsymbol{\theta}) - \lambda_i \nabla_{\theta_i} \left[ J_i(\theta_i'(\boldsymbol{\theta}), \theta_{-i}) - J_i(\boldsymbol{\theta}) \right], \quad (9)$$

where $\theta_i'(\boldsymbol{\theta})$ is the parameters of the optimal unilateral deviation $\pi_i^*$ for agent $i$, i.e. the parameters that minimize local regret, which depends on the current policy parameters $\theta_i$. $\lambda_i$ is updated as:

$$\nabla_{\lambda_i} J_{p,\min}^{\dagger}(\pi_p, \epsilon) = J_i(\pi_i^*, \pi_{-i}, \pi_p) - J_i(\boldsymbol{\pi}, \pi_p) - \epsilon_i. \quad (10)$$

Equation 8 still poses a challenge through the $J_i(\pi_i^*, \pi_{-i}, \pi_p)$ terms, which correspond to unknown agent regret. We now detail the efficient approximation of the value and derivative of agent regret using local and meta-learning approximations, respectively.

**Trust Regions using Local $\epsilon$-equilibria**  Estimating regret requires knowledge of the optimal unilateral deviation for agent $i$. We can simplify this problem by proposing a refinement of $\epsilon$-equilibria inspired by the notion of local Nash equilibria in differentiable games (Ratliff et al., 2014).

**Definition 3.1.** A strategy $\boldsymbol{\pi}$ is a local $\epsilon$-Nash equilibrium if there exists open sets $W_i \subset \Pi^N$ such that $\pi_i \in W_i$ and for each $i \in \{1, \ldots, N\}$ we have that $J_i(\pi_i', \pi_{-i}) \leq J_i(\boldsymbol{\pi}) + \epsilon'$ for all $\pi_i' \in W_i \setminus \{\pi_i\}$, where $\epsilon' := \epsilon \sup_{\pi_i' \in W_i} KL(\pi_i || \pi_i')$.

By instead performing `Agent-Adv-Search` on the local $\epsilon$-Nash equilibria, we can limit the set of unilateral deviations to consider to a small trust region, $\Pi_\eta(\pi)$:

$$\text{ANE}(\pi_p, \eta) := \{ \boldsymbol{\pi} \mid \forall i \in [1, N], \tilde{\pi}_i \in \Pi_\eta(\pi_i) : J_i(\tilde{\pi}_i, \pi_{-i}, \pi_p) \leq J_i(\pi_i, \pi_{-i}, \pi_p) + \epsilon \}, \quad (11)$$

$$\Pi_\eta(\pi) := \{ \pi' \in \Pi \mid KL(\pi || \pi') \leq \eta \}, \quad (12)$$

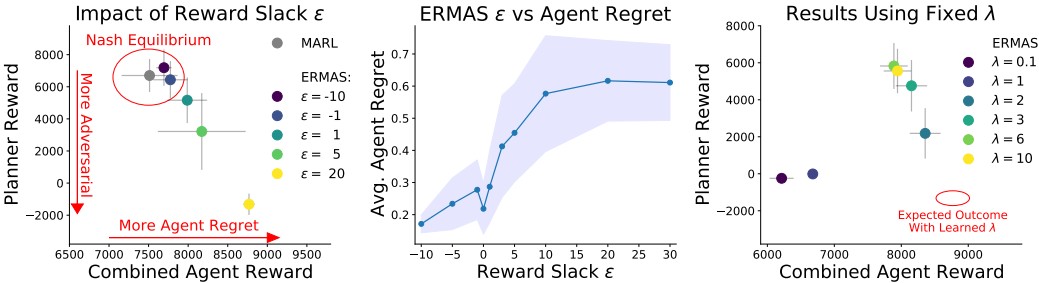

Figure 2: Validating ERMAS `Agent-Adv-Search` in constrained repeated bimatrix games. Each point in the above scatter plots describes the average outcome at the end of training for the agents ($x$-coordinate) and the planner ($y$-coordinate). Error bars indicate standard deviation. **(Left)** The bimatrix reward structure encodes a social dilemma featuring a low-agent-reward/high-planner-reward Nash equilibrium, which is where vanilla MARL converges. Agents trained with ERMAS deviate from this equilibrium in order to reduce planner reward. $\epsilon$ governs the extent of the allowable deviation. **(Middle)** As $\epsilon$ increases, the average per-timestep regret experienced by the agents also increases. Each average is taken over the final 12 episodes after rewards have converged. **(Right)** Using fixed values of $\lambda$ (rather than allowing it to update, as in the full algorithm) distorts performance and prevents agents from reaching the same $\epsilon$-equilibria discovered with learned $\lambda$.

where $\eta > 0$ defines the size of the trust region. For small $\eta$, algorithms such as TRPO (Schulman et al., 2017) can be used to efficiently approximate optimal local deviations of $\pi_i$, affording reasonable approximations of $J_i(\pi_i^*, \pi_{-i}, \pi_p)$. Note that our usage of trust region algorithms is not for optimization purposes. ERMAS requires the use of trust region optimization to ensure that the equilibria considered by ERMAS are limited to a local neighborhood of the policy space (Eq 11).

**First-Order Meta Learning Approximation**    The full gradient in Equation 9 is also complicated by the need to estimate the derivative of local regret $\nabla_{\theta_i} [J_i(\theta_i'(\boldsymbol{\theta}), \theta_{-i}) - J_i(\boldsymbol{\theta})]$. The second term maximizes the performance of the agent's policy and is simply found with policy gradient. The first term is less straightforward: it minimizes the performance of the best agent policy in the current trust region. We note that this first term corresponds to a meta-learning gradient. We follow REPTILE (Nichol et al., 2018) to obtain a first-order approximation of a $M$-step meta-learning gradient:

$$\nabla_{\theta_i} J_i(\theta_i'(\boldsymbol{\theta}), \theta_{-i}) = g_1 - \frac{1}{M} \sum_{i=1}^{M} g_i, \quad g_i = \nabla_{\theta_i} J_i\left(\theta_i + \sum_{j=1}^{i-1} g_j, \theta_{-i}, \theta_p\right), \quad (13)$$

where $g_i$ denotes the $i$th policy gradient in the direction of $J_i$. In practice, we scale this meta-learning term with the hyperparameter $\beta$ as $\beta < 1$ incorporates a helpful inductive bias where maximizing agent reward leads to local maxima. We can alternatively apply this gradient update periodically, to both mimic $\beta < 1$ and reduce computation overhead. First-order meta-learning approximations are known to be empirically effective, and are necessary for ERMAS to efficiently solve Eq 8.

**ERMAS**    By solving the dual problem (Eq. 8), ERMAS yields robustness to $\epsilon$-equilibria and, equivalently, uncertainty in agent objectives. ERMAS solves the dual problem by combining trust regions and meta-learning techniques to estimate and differentiate agent regret. Algorithm 1 and 2 (Appendix A.1) describe an efficient implementation of this procedure for nested policy learning.

## 4    EXPERIMENTAL VALIDATION IN MULTI-AGENT SIMULATIONS

### 4.1    ERMAS SOLVES THE INNER LOOP: AGENT-ADV-SEARCH

**Constrained Repeated Bimatrix Game**    We analyze the agent behaviors learned by `Agent-Adv-Search` in experiments depicted in Figure 2. These experiments apply ERMAS to the classic *repeated bimatrix game*, which is well-studied in the game theory literature as its Nash equilibria can be solved efficiently. At each timestep, a row player (Agent 1) and column player

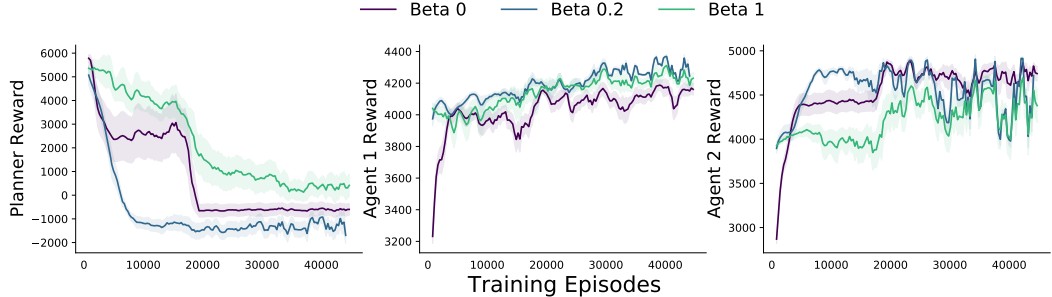

Figure 3: Planner and agent rewards in the repeated bimatrix game over training time. Each line represents an average over 10 seeds, with error bars indicating standard error. The lines correspond to runs of ERMAS and are colored according to $\beta$, the weight of the meta-learning term of Equation 13. **(Left)** Planner performance over training episodes. **(Middle)** Agent 1 performance over training episodes. **(Right)** Agent 2 performance over training episodes.

(Agent 2) simultaneously choose from a finite number of actions. Each pair of actions $(i, j)$ corresponds to a pair of payoffs for the agents $r_1(i, j), r_2(i, j)$. We select the payoff matrices $r_1$ and $r_2$, illustrated in Figure 1a, so that only one Nash equilibrium exists and that the equilibrium constitutes a "tragedy-of-the-commons," where agents selfishly optimizing their own reward leads to less reward overall. To extend this *repeated bimatrix game* into sequential decision making, we further constrain the game so that, at any timestep, agents can only choose an action adjacent to their action in the previous timestep. We also introduce a passive planner that observes the game and receives a payoff $r_p(i, j)$. The planner does not take any actions and its payoff is constructed such that its reward is high when the agents are at the Nash equilibrium. In effect, this toy setting allows us to verify that ERMAS samples realistic worst-case behaviors–that is, that Agents 1 and 2 learn to deviate from their tragedy-of-commons equilibrium in order to reduce the planner's reward but also without significantly increasing their own regret.

**Discovering $\epsilon$-Equilibria**  Figure 2 (left) visualizes how the equilibria reached by AI agents balance the reward of the planner (y-axis) and agents (x-axis). Conventional multi-agent RL discovers the Nash equilibrium, which is visualized in the top left. At this equilibrium, the agents do not cooperate and the planner receives high reward. For small values of $\epsilon$, ERMAS also discovers the Nash equilibrium. Because $\epsilon$ acts as a constraint on agent regret, larger values of $\epsilon$ enable ERMAS to deviate farther from the Nash equilibrium, discovering $\epsilon$-equilibria to the bottom-right that result in lower planner rewards. Deviations from a Nash equilibrium are associated with higher regret, meaning that regret should increase with $\epsilon$. Figure 2 (middle) clearly demonstrates that ERMAS imparts this trend. As described by Equation 7, this display of adversarial behavior is key to learning a more robust planner.

## 4.2 All the Components of ERMAS are Necessary

**Dynamic or Frozen Lagrange multipliers $\lambda_i$**  The Lagrange multipliers $\boldsymbol{\lambda}$ balance the dual objectives of seeking stable agent equilibria and minimizing the planner's reward. Recall that smaller values of $\lambda$ mean agent objectives are more antagonistic. The $\boldsymbol{\lambda}$ are updated using local estimates of agent regret, as described in Equation 8. We can validate that these updates are necessary by analyzing the equilibria learned by ERMAS when $\boldsymbol{\lambda}$ are not updated. Fixing a value of $\boldsymbol{\lambda}$ reduces Equation 8 to learning with shared rewards. Figure 2 (right) visualizes the equilibria discovered with frozen $\boldsymbol{\lambda}$ in the same format as the Figure 2 (left) plot. This visualization shows that freezing $\boldsymbol{\lambda}$ affects the equilibria discovered by ERMAS; the bottom right quadrant which contains the $\epsilon$-equilibria discovered with dynamic $\lambda$ are not reached for any values of $\lambda$. This validates that certain $\epsilon$-equilibria are only reachable with dynamic $\lambda$ and hence the updates in 8 are necessary for proper behavior.[4]

---

[4]However, we recommend temporarily freezing $\lambda$ at the top of Algorithm 2 to "warm up" agent policies.

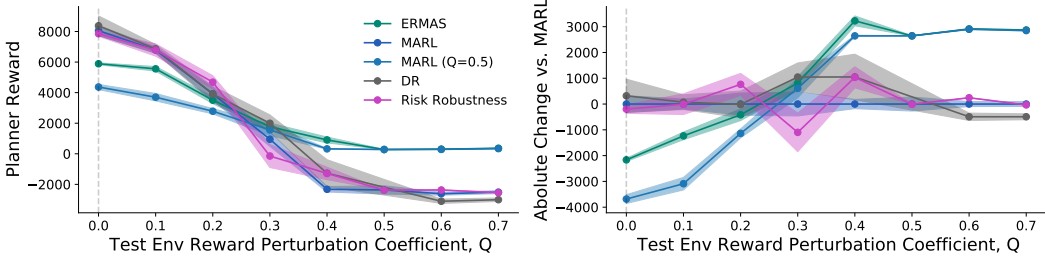

Figure 4: Robust planner learning in the repeated bimatrix game. Average planner performance as a function of the reward perturbation coefficient $Q$ in the test environment. Each point represents an average over 5 seeds, with error bars indicating standard error. Vertical dashed lines denote the training environment $Q$. **(Left)** Average planner performance at convergence. **(Right)** Performance relative to the vanilla MARL baseline, which is trained without a robustness objective.

**First-order Approximation**  Using the first order approximation of Equation 13 might slow learning, since we step "away" from the direction that the standard policy gradient suggests. In contrast, we find that the ERMAS update can significantly speed up convergence. Figure 3 depicts the learning curve of variants of ERMAS with varying weights, $\beta$, on the meta-learning term. For instance, when $\beta = 0$, the meta-learning term is completely dropped. We find that lines corresponding to larger values of $\beta$ reach improved agent rewards half an order of magnitude faster than lower values of $\beta$. However, larger values of $\beta$ fail to reach the desirable region of adversarial $\epsilon$-equilibria. In both cases, very large ($\beta > \frac{1}{N}$) and very small ($\beta \sim 0$) fail to fully reach and remain at the adversarial region. However, an appropriate range for $\beta$ gives rapid convergence and good asymptotic behavior.

## 4.3    SOLVING PLANNER-OPT USING AGENT-ADV-SEARCH

We now show that ERMAS yields planners $\pi_p$ which are more robust to uncertainty in the agent's reward function. Specifically, we empirically validate that ERMAS can find strong solutions to the nested optimization problems Agent-Adv-Search and Planner-OPT, in two simulations: repeated bimatrix games and economic simulations (see Figure 1).

**Evaluation Workflow**  To measure planner robustness, we evaluate trained planners in test environments with a reality gap (i.e. reward parameterizations that differ from the training environment), containing test agents that are optimized for the test environment's reward functions and that have not been seen by the planner. We proceed as follows: 1) Train the agents and planner with ERMAS. 2) Fix the planner and transfer it to a test environment, then train new agents *from scratch* in the test environment with reward functions unseen by the planner. 3) Report the value of the planner objective after the new agents have converged.

**Augmented Repeated Bimatrix Games**  We now extend the bimatrix game to a nested reinforcement learning problem by allowing the planner to take actions and influence its own experienced reward. The planner now selects an integer $a \geq 0$ that modifies its new reward function: $ar_p(i,j) - a^2$. This function is chosen for its quadratic form: $a$ scales the reward but adds a cost $a^2$ which disincentives large values of $a$. Agents are modified to receive a new reward function $r_i' = r_i - Qr_p$ where $r_i$ is their original reward function and $Q \leq 1$ is an unknown scalar which may differ between training and testing environments. If $Q = 0$, agents act identical to the experiments in Figure 2. For environments with larger values of $Q$, e.g., $Q = 0.5$, agents have an incentive to be adversarial.

Figure 4 shows the test performance of ERMAS against vanilla multi-agent RL policies learned with $Q_{\text{train}} = 0.5$ or with $Q_{\text{train}} = 0$. Naturally, MARL trained with $Q_{\text{train}} = 0.5$ underperforms MARL trained with $Q_{\text{train}} = 0$ when evaluated in environments where $Q_{\text{test}} < 0.3$. However, MARL trained with $Q_{\text{train}} = 0.5$ outperforms when $Q_{\text{test}} > 0.3$. Agent-Adv-Search successfully finds the worst-case perturbation, and yields a planner policy at least as good as MARL $Q_{\text{train}} = 0.5$. This shows that ERMAS can produce policies which are robust to uncertainty in agents' objectives and is a tractable adversarial robustness algorithm.

Existing single-agent and multi-agent robust reinforcement learning algorithms optimize for different robustness objectives than ERMAS. However, some general techniques for addressing Sim2Real can be extended to provide an alternative to ERMAS's adversarial approach. Our first baseline extends the technique of domain randomization (DR) by applying random perturbations to agent rewards. For DR, agents receive $r'_i = r_i + \sigma$, where $\sigma \sim U[-1, 1]$ (recall $r_i \in [0, 1]$). These perturbations are randomized periodically between episodes. Even in a simple 4-by-4 bimatrix game, there are $2304 = 16$ states $\times 16$ states $\times 9$ actions possible reward values to perturb; a 2304-dimensional space is too sparse to cover uniformly. Furthermore, in contrast to DR of visual inputs or dynamics, there is a natural latency to the effect of DR of agent rewards: randomization only has an effect if agents learn to adapt to their perturbed rewards. Our second baseline, risk robustness (RR), applies a concavity to planner rewards, e.g., $\log r_p$. This encourages robustness to environment stochasticity, including the randomness of agent policies. As illustrated in Figure 4, neither the domain randomization nor risk robustness baseline yields meaningful robustness even in this simple setting.

### 4.4 ERMAS FOR LARGE-SCALE TWO-LEVEL LEARNING: AI ECONOMIST

We now address an important use-case, tax policy design, to demonstrate ERMAS at scale. We train a robust AI Economist: a planner for optimal taxation that acts in a spatiotemporal economic simulation, see Zheng et al. (2020) for a detailed description. This setting is very challenging given the scale of the simulation and agent affordances. In particular, *adversarial and randomization baselines do not scale to this setting.* In this simulation, agents optimize their expected utility $\mathbb{E}_{\boldsymbol{\pi}, \pi_p}\left[\sum_{t=0}^{T} r_{i,t}\right]$, where the utility $r_{i,t}$ is a function of labor $l_{i,t}$ and post-tax endowments $\tilde{x}_{i,t}$:

$$\tilde{z}_{i,t} = z_{i,t} - T(z_{i,t}), \quad \tilde{x}_{i,t} = \sum_{t' \leq t} \tilde{z}_{i,t}, \quad r_{i,t}(\tilde{x}_{i,t}, l_{i,t}) = \frac{\tilde{x}_{i,t}^{1-\eta} - 1}{1 - \eta} - l_{i,t}, \quad \eta > 0, \quad (14)$$

where $\eta$ sets the degree of risk aversion (higher $\eta$ means higher risk aversion). Agents earn income $z_{i,t}$ and pay taxes $T(z_{i,t})$, which are set by the planner. The planner optimizes social welfare `swf`:

$$\texttt{swf} = \texttt{eq}(x) \cdot \texttt{prod}(x), \quad \texttt{eq}(x) = 1 - \frac{N}{N-1}\texttt{gini}(x), \quad \texttt{prod}(x) = \sum_{i=1}^{N} x_i, \quad (15)$$

a combination of equality (Gini, 1912) and productivity.

Figure 5 shows the performance of ERMAS tax policies along with baseline tax policies (Saez, US Federal) and an AI Economist (MARL). All models are trained on agents with $\eta = 0.23$. We replicate Zheng et al. (2020): MARL outperforms the Saez tax. However, MARL fails to outperform Saez when testing with $\eta < 0.22$ and $\eta > 0.28$. In contrast, ERMAS yields consistent gains across the depicted range of $\eta$. This shows that ERMAS's tax policy can improve `swf` even if agent risk aversion significantly increases or decreases (e.g., due to economic cycles or exogenous shocks).

In fact, ERMAS outperforms the baseline AI Economist for the original setting of $\eta = 0.23$. This suggests the robustness and performance do not necessarily pose a zero-sum game: ERMAS can find equilibria with high performance and strong generalization. It has been observed empirically in various studies that single-agent robust reinforcement learning similarly yields performance improvements even in the absence of environment perturbations (Pinto et al., 2017).

## 5 RELATED WORK

**Sim2real: Reality Gaps**  The Sim2Real problem considers how reality gaps, i.e., mismatches between a simulation and reality, can (negatively) impact RL policies. Robust performance across reality gaps has been studied when applying deep RL to robotic control with visual inputs (James et al., 2019; Christiano et al., 2016; Rusu et al., 2016; Tzeng et al., 2015). To close reality gaps, domain adaptation aims to transfer a simulated distribution (of the world) to the real world (Higgins et al., 2017; Kim et al., 2019). In comparison, the study of reality gaps in multi-agent RL is relatively nascent (Suh et al., 2019; Nachum et al., 2019).

**Multi-Agent Robustness**  Morimoto & Doya (2001) proposed robust RL for the worst-case adversarial objective (i.e., a max-min optimization problem), inspired by $\mathcal{H}_\infty$ control, and provided

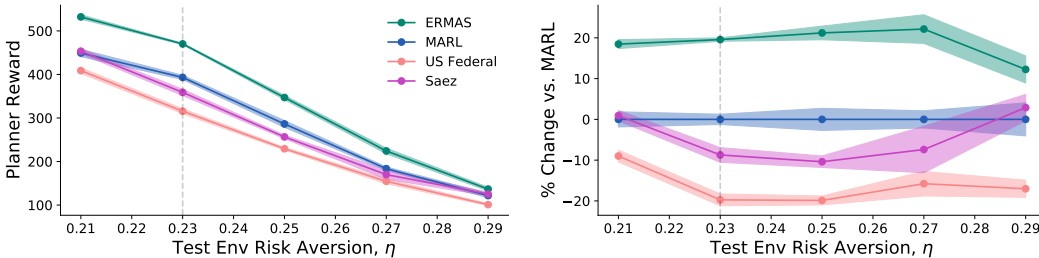

Figure 5: Learning robust optimal taxation in an economic simulation. Results are plotted following the same conventions in Figure 4. Due to the intractability of other robustness algorithms in this setting, we instead compare against different planner choices. "US Federal" uses a fixed tax scheme adapted from the 2018 US Federal income tax rates. "Saez" uses an adaptive, theoretical formula to estimate optimal tax rates.

analytical solutions in the linear setting. The robust RL problem can be solved using an adversarial player that chooses perturbations. Rather than the worst-case, Pinto et al. (2017) optimized for the $\alpha$-quantile of reward, using a conditional value-at-risk (CVAR) interpretation of adversarial robustness. We consider perturbations to agent rewards. Other works have studied perturbing transition matrices, observation spaces, and action probabilities, which are relevant to robotics and control applications (Pinto et al., 2017; Tessler et al., 2019; Hou et al., 2020; Li et al., 2019). We also address a more challenging multi-agent setting than previous works. First, agents do not necessarily optimize for the same (robustness) objective. We therefore consider a *one-sided robustness* problem where only the planner is expected to act robustly. This setting is significantly more difficult than previous works that assume all agents act robustly (Li et al., 2019). Second, we address a challenging nested reinforcement learning setting of interest to important real-world applications such as economics mechanism design (Zheng et al., 2020). Although we formulate ERMAS as using simultaneous gradient descent, ERMAS can be easily extended to use nested optimization solvers such as competitive gradient descent (Schäfer & Anandkumar, 2020).

See Appendix A.2 for additional related work.

## 6 FUTURE WORK

Future work could extend ERMAS to reality gaps due to different causes, such as changes in transition dynamics, agent affordances, number of agents, etc, which might require more refined approximations to the planner's robustness objective. Furthermore, it would be interesting to apply ERMAS to problems that feature more general solution concepts, e.g., Bayesian Nash equilibria, correlated equilibria, Walrassian equilibria, and others.

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
