# OpenReview forum: "ERMAS: Learning Policies Robust to Reality Gaps in Multi-Agent Simulations"
_ICLR.cc/2021/Conference — Reject_

### Official Review · AnonReviewer1 · 2020-10-26
**Great motivation, some minor qualms on execution**

**Rating:** 7
**Confidence:** 3

**Review:**

This paper proposes a method to learn _robust_ policies in multiagent environments: in particular, the policies should continue to work even if other agents in the environment deviate slightly in their behavior. Unlike previous work which allows for perturbations in other agents’ policies, this work allows for perturbations in the reward functions that those agents optimize, allowing for more relevant robustness.

Overall I liked the work -- robustness to what the agents optimize seems quite important and (to my limited knowledge) novel. The experiments seem reasonable and establish that the method (ERMAS) works well. I note a few qualms below.

----

One worry about the theory is that it is not well-defined because it allows for arbitrary exogenous perturbations, as in this line:

> The robustness objective in Equation 4 considers general agent perturbations, e.g., agents may exhibit an exogenous aversion to complex policy choices.

With this, it is no longer necessarily the case that a Nash equilibrium exists: Nash equilibria are only guaranteed to exist when the utility functions are of the _game outcomes_; they need not exist when the utility functions can also apply to the _chosen policies_.

For example, consider rock-paper-scissors between Alice and Bob. Wins get +1 utility, losses get -1, and ties get 0. Now suppose we add the perturbation “-100 if you play a stochastic policy” (normally not allowed, but allowed in this paper’s formalism as an “exogenous preference”). It is easy to see that there is no Nash equilibrium where either player plays a stochastic policy, as they could then switch to a deterministic policy. But we know there is no Nash equilibrium for rock-paper-scissors under deterministic policies.

Since Nash equilibriums are no longer guaranteed to exist, the objective in (4) may be undefined in some settings.

This could be fixed by restricting perturbations to only affect utilities of states, though this would significantly decrease the expressive power of perturbations.

----

I was confused by the fact that ERMAS outperformed MARL even when there is no change in agent policies. The authors note this fact as well:

> In fact, ERMAS outperforms the baseline AI Economist for the original setting of η = 0.23. This
suggests the robustness and performance do not necessarily pose a zero-sum game: ERMAS can find equilibria with high performance and strong generalization.

I am not sure I share the authors’ enthusiasm. It is straightforward to show that the optimal policy in the MARL setting does at least as well as the optimal policy in the robust setting. The fact that ERMAS actually outperforms MARL is surprising and suggests that ERMAS optimizes better than MARL for some reason. This could happen in one of two ways:

1. Something about ERMAS leads to better learning, in a way that can’t apply to MARL (e.g. perhaps by having adversaries learning progresses faster since any flaws are found more quickly)
2. Something about ERMAS leads to better learning, in a way that could apply to MARL (e.g. better hyperparameter tuning).

If it’s the second case (especially hyperparameter tuning), then it calls into question whether any of the improvements in the experiments come from the design of ERMAS, rather than coming from something unrelated like better hyperparameter tuning.

----

Quality: Decent. I liked the algorithm derivation (though I did not carefully check the math), and the evaluation does show the benefits of the approach, though it would have been nice to test the approach on more environments (there is just one toy environment and one more complex environment). In particular, as far as I can tell the algorithm can be applied to any multiagent RL (MARL) problem; there are several benchmark suites on which this could be tested.

Clarity: I found the paper to be quite clear.

Originality: I believe this is a novel formulation, though I am not very familiar with the robust RL literature.

Significance: Clearly relevant and important.

---

> ### Author Response · Authors · 2020-11-18
> **Response to Reviewer 1**
>
> We thank the reviewer for the detailed feedback.
>
> *“One worry about the theory is that it is not well-defined because it allows for arbitrary exogenous perturbations, as in this line:”*
> The perturbations we consider are meant to be free from the Markov assumption so that perturbations can depend on an agent's entire trajectory rather than only the most recent state. We did not intend for perturbations to freely depend on the policy space, where a  perturbation could penalize stochasticity, as you noted. We have corrected this in the revision: the perturbation $\xi$ should only depend on a policy in terms of the expected trajectories. This remedies the theoretical issue and addresses your concern about the well-definedness of Equation 4. We note that this change is a technical correction in the motivational exposition; it does not affect any derivations or the proposed algorithm. We thank you for catching this.
>
> *“I was confused by the fact that ERMAS outperformed MARL even when there is no change in agent policies.”*
> Robust reinforcement learning often yields policies that perform better even in the absence of perturbations/disturbances, especially in complex learning problems. This behavior is well-known and noted by seminal literature (see Pinto et al. 2017). We point this out in our writing to highlight that ERMAS yields the benefits expected from existing robust RL techniques; we have also revised the paper to make this point clearer. We do not believe that these improvements reflect hyperparameter tuning. With the exception of hyperparameters that are unique to ERMAS (e.g., trust region size eta, reward slack epsilon), all hyperparameters used in ERMAS (e.g., learning-rate, batch-size) are equivalent to those used in the baselines, which are the same used in previous literature: see Table 2 in Zheng et al. 2020. Our baselines replicate the performance reported in that paper.
>
> *“...the evaluation does show the benefits of the approach, though it would have been nice to test the approach on more environments”*
> We agree that ERMAS is general and can be applied to a broad range of problems. In this paper, we chose to focus on an intuitive use-case with immediate importance, tax policy design, to demonstrate the algorithm’s performance and behavior. We also point out that ERMAS considers one-sided robustness (as discussed in the Section 5), which applies more naturally to our demonstrated settings than to standard MARL benchmarks.
>
> L. Pinto, J. Davidson, R. Sukthankar, and A. Gupta, “Robust Adversarial Reinforcement Learning,” arXiv:1703.02702 [cs], Mar. 2017, Accessed: Nov. 17, 2020. [Online]. Available: http://arxiv.org/abs/1703.02702.
>
> S. Zheng, A. Trott, S. Srinivasa, N. Naik, M. Gruesbeck, DC. Parkes, and R. Socher, “The AI Economist:  Improving Equality and Productivity with AI-DrivenTax Policies,” arXiv:2004.13332,  April 2020. http://arxiv.org/abs/2004.13332.

---

> > ### Comment · AnonReviewer1 · 2020-11-25
> > **Thanks for the response**
> >
> > I don't have any other questions at this time. I do still think testing on more environments would significantly strengthen the paper, but the current results do seem reasonable. I'm raising my score to 7.

---

### Official Review · AnonReviewer4 · 2020-10-28
**weakly reject**

**Rating:** 6
**Confidence:** 3

**Review:**


##########################################################################

Summary:
This paper purposes a robust optimization framework for training robust policies, such that policies learned in imperfect
simulations can be robust against reality gaps in order to be effective in the real world. The multi-agent adversarial robustness problem is derived using \eqsilon-Nash equilibria; the paper purposes ERMAS, and present how ERMAS solves the nested robust optimization problem using dualization, trustregions, and first-order meta-learning techniques; the authors  empirically validate ERMAS by training robust policies in two multi-agent problems: sequential bimatrix games and economic simulations.

##########################################################################

Reasons for score:

1. Overall I think the reality gap is a very important problem, and the paper is well organized and presented clearly. The robust optimization formulation provides a way to approach this problem. I have several concerns about this formulation (stated below)

2. The robust optimization formulation requires an input of the size of the uncertainty set. However I did not find discussion in the paper on how such a bound on the uncertainty set can be obtained or estimated in practice. If such a bound is loose, the robust optimization approach might become overly conservative. Following on this conservativeness concern, ERMAS learns the robust policies by finding the worst-cas reality gaps, which corresponding to highly adversarial agents --so a small fraction of highly adversarial agents could be sufficient make the uncertainty set size large.

3. I would like to see more discussions on the approximate feasibility and optimality for solving the robust optimization problem eq(7).

4. In the experiments (figure 2), it is unclear that if the average per-time step regret experienced by the agents increases as the reward slack \epsilon increases (the trend becomes decreasing as \epsilon is large)


##########################################################################

Questions during rebuttal period:


Please address and clarify the cons above


#########################################################################

--post discussion--
The authors' response are helpful in addressing the concerns in the original review. I decided to update the score to 6.

---

> ### Author Response · Authors · 2020-11-18
> **Response to Reviewer 4**
>
> We thank the reviewer for the detailed feedback.
>
> *“However I did not find discussion in the paper on how such a bound on the uncertainty set can be obtained or estimated in practice.”*
> We will add a discussion on how to bound uncertainty sets at the end of Section 2. ERMAS uses uncertainty sets that measure uncertainty about the objectives of other agents. The “size” of this uncertainty set is given by $\epsilon$: the amount by which the reward functions of other agents can differ between the train and test environments, e.g., between simulation and the real world. This quantity can be approximated with domain knowledge about the expected error of agent reward objective estimates. However, in general, bounding the size of the uncertainty set is a hyperparameter that should be tuned; this is also the case with other existing adversarially robust RL algorithms.
>
> *“Following on this conservativeness concern, ERMAS learns the robust policies by finding the worst-cas reality gaps, which corresponding to highly adversarial agents --so a small fraction of highly adversarial agents could be sufficient make the uncertainty set size large.”*
> We agree with the reviewer’s comment that “a small fraction of highly adversarial agents could be sufficient to make the uncertainty set size large.” It’s possible to train only a subset of the agents with the adversarial objective, but this is outside the scope of our current work.
>
> *“I would like to see more discussions on the approximate feasibility and optimality for solving the robust optimization problem eq(7).”*
> Feasibility: The constraint set is non-empty for $\epsilon \geq 0$. Our constraints enforce an upper bound $\epsilon$ on the regret of agents. By definition, in a non-empty policy space, there must exist an optimal policy with zero regret.
> Optimality: ERMAS has three sources of approximation error: first-order approximation of a meta-learning objective (Eq 13), the duality gap associated with Eq 9, and normal learning error. First-order meta-learning approximations are broadly accepted as reasonably good, and are unavoidable in our problem (Nichol et al. 2018). Our duality gap is the same duality gap which arises from dual solutions to constrained reinforcement learning. Such duality gaps are known to be negligible in practice and provably zero under weak assumptions (Paternain et al. 2019).
> We will revise the paper to add this commentary.
>
> *“Figure 2, it is unclear that if the average per-time step regret experienced by the agents increases as the reward slack \epsilon increases (the trend becomes decreasing as \epsilon is large)”*
> The appearance of a reversed trend for large values of $\epsilon$ are simply due to noise (reflected by the error bars). Intuitively, when agents optimize their objective (i.e. aim to reduce regret), estimates of regret are more reliable than when agents optimize a different objective (i.e. don’t aim to reduce regret). As $\epsilon$ increases, agents have more “slack” to be adversarial rather than optimize their objective. This means large $\epsilon$ values tend to produce noisier regret measurements (see the error bars). To make the trend more clear, we have revised the Figure 2 middle plot to average measurements over a larger window of time: the last 33% of training episodes (approximately 12 episodes per random seeds, for 10 seeds per epsilon). The trend of regret increasing with $\epsilon$ should now be more visually evident.
>
> A. Nichol, J. Achiam, and J. Schulman, “On First-Order Meta-Learning Algorithms,” arXiv:1803.02999 [cs], Oct. 2018, Accessed: Nov. 17, 2020. [Online]. Available: http://arxiv.org/abs/1803.02999.
>
> S. Paternain, L. F. O. Chamon, M. Calvo-Fullana, and A. Ribeiro, “Constrained Reinforcement Learning Has Zero Duality Gap,” arXiv:1910.13393 [cs, math, stat], Oct. 2019, Accessed: Nov. 17, 2020. [Online]. Available: http://arxiv.org/abs/1910.13393.

---

### Official Review · AnonReviewer3 · 2020-10-28
**Robust RL under the multi-agent setting.**

**Rating:** 6
**Confidence:** 1

**Review:**

Summary:

This paper tackles robust RL under the multi-agent setting. They formulate the multi-agent adversarial robustness problem as a nested optimization problem and propose a practical algorithm (ERMAS) to solve it. Theoretical proof and empirical study on two environments are provided to demonstrate the effectiveness of the proposed framework.

##########################################################################

pros:

+ The paper is well written with clear and interesting motivation. Bridging the gap between simulation and reality for multi-agent environments is an important topic. Although I'm not an expert in this direction, it feels to me the proposed framework is technically and theoretically sound with comprehensive empirical evaluations.

##########################################################################

cons:

- It would be better to include more baselines from the MARL field, especially the ones you cited in related work that also considers robust MARL. Only a vanilla MARL baseline cannot actually tell how much you improve empirically from SOTA.
- It would be better to include literature from the nested optimization field in the related work. Since you formulate the robust MARL as a nested optimization problem, have you considered or compared ERMAS with any other existing nested optimization solvers?


##########################################################################

Post Rebuttal

The paper has been updated to include additional reviews about nested optimization. I would like to keep my original score.

---

> ### Author Response · Authors · 2020-11-18
> **Response to Reviewer 3**
>
> We thank the reviewer for the detailed feedback.
>
> *“It would be better to include more baselines from the MARL field, especially the ones you cited in related work that also considers robust MARL. Only a vanilla MARL baseline cannot actually tell how much you improve empirically from SOTA.”*
> There are two main issues in applying baselines: (1) applicability, and (2) scalability. Most baselines do not apply to our problem because we are addressing a novel robustness objective. The robustness literature has proposed algorithms that optimize for different robustness objectives, for example, robustness to perturbations in agent observations/actions/environment transitions. In order to provide baselines for our robustness objective, we apply the well-known techniques of domain randomization (apply random perturbations) and risk robustness (apply concavity to the robust agent’s reward). These techniques are popular in the Sim2Real literature and generalize to different robustness objectives, including ours. These baselines are seen in Figure 4.
> In the economic simulation environment, the domain randomization baseline does not scale. Domain Randomization of reward functions prevents meaningful learning in large state spaces as the resulting noise significantly destabilizes learning.
>
> *“It would be better to include literature from the nested optimization field in the related work. Since you formulate the robust MARL as a nested optimization problem, have you considered or compared ERMAS with any other existing nested optimization solvers?”*
> ERMAS is not an alternative/competitor to nested optimization. ERMAS uses simultaneous gradient descent (as is common practice in deep RL) but can be extended to use nested optimization solvers such as competitive gradient descent (Schafer et al. 2019). We plan to address this literature in revisions to the Related Works section.

---

### Official Review · AnonReviewer5 · 2020-11-06
**Very technical and needs more motivation**

**Rating:** 6
**Confidence:** 3

**Review:**

This paper proposes an interesting method for being able to act and plan robustly in a multiagent simulation and be robust to the reality gap between training time and testing time for agents in a marl setting. the method does show improvements in terms of being able to train the policy for this use case and being more robust to some out of distribution configuration of the environment however these improvements appear to be rather limited. in addition, the organization and writing for the paper is very technical and could be improved with additional background information on the uses of metrics and environments as well as better flow between the content in the paper to understand the importance of the different aspects of the method. These improvements could help the reader understand the novelty and important aspects of the method that are difficult to measure. At the moment it comes across as a mix of different methods combined to be able to support this more robust method without a very clear story about the primary problem the paper is trying to solve or the more significant technical aspect of the method that provides this novel solution.

The paper appears to be very technical. The impact of this paper on a more general audience is going to be limited due to the number of details left out of the paper that would help motivate the reasoning for using certain metrics for learning why you would want bounded regret and Nash equilibria. For example, in figure 2 all of the plots have different quantities on the x and y-axis that are only briefly described and not very well motivated why these are good metrics for analyzing this type of algorithm. The writing really needs to include more motivation to clearly describe why these environments and metrics are the right kind of analysis to perform in order to indicate the efficacy of the method.

For example they bi Matrix game used at the beginning of the experiment section has no explanation. How does this game work how can we understand whether or not the method is improving in this game without understanding the difficulties and or nuanced importance of analyzing this environment.

In the augmented repeated bi Matrix game scenario that is used in test environment that is a modified version of the bi matrix game in order to show the robustness of the method there, is again, not a clear description about why these environment changes would be significant. Overall since little is understood about this game one can only assume that these particular changes to the environment could be selectively biased in order to favor the method that's proposed in the paper and overall the analysis of this section is unclear.

Figure four does not appear to be referenced in the paper anywhere I'm going to assume that it should be referenced inside of the augmented repeated by metrics game section and if that is the case the marl baseline is not very well explained for this figure before it is used in figure 4.

Additional Comments:
- The list of contributions at the end of the introduction seems to be fairly repetitive from the content in the last paragraph in the introduction.
- The part the connects the work to TRPO is not discussed carefully paper. Is the method just using TRPO for multi-agent RL?
- In figure 3 we will assume that the ordering in the text is actually left center-right.


----- Post Discussion ----
Updates to the paper have helped make technical parts of the paper more clear. The paper has also been edited to improve the motivation and experimental explanation.

---

> ### Author Response · Authors · 2020-11-18
> **Response to Reviewer 5**
>
> Thank you for your valuable feedback and for pointing out ways to improve our writing in your “additional comments”. Since most of your comments concerned the motivation of our work, please see the high-level summary in the general response to all reviewers. We hope this provides adequate context for our work. We agree that the paper can more clearly motivate and explain the environments and experimental metrics used, and how they are used to validate ERMAS. We will revise the text accordingly.
>
> *“For example they bi Matrix game used at the beginning of the experiment section has no explanation. How does this game work how can we understand”*
> We extend the classic bimatrix game into a repeated bimatrix game (2 agents) and a repeated trimatrix game (2 agents + 1 planner). This is a fast and interpretable environment where we can cleanly analyze ERMAS. Repeated bimatrix games are well-studied in the game theory literature as bimatrix game Nash equilibria (if they exist) can be explicitly solved for. For example, our analysis leverages the social dilemma structure of Nash equilibria in this game, in order to clearly verify that ERMAS can train non-robust agents to act adversarially to train the robust agent, as depicted in Figure 2. Specifically, non-robust agents discover the Nash equilibrium and deviate from it adversarially to an extent that we can control with the hyperparameter epsilon.
>
> *“Is the method just using TRPO for multi-agent RL?”*
> We do not simply use TRPO as an MARL algorithm; the ERMAS objective has a deeper connection with trust regions and meta-learning. We discuss the relationship between ERMAS and trust regions (not necessarily TRPO) in the paragraph containing Equation 11-12 and also in our main summary response. ERMAS requires information about the optimality of policies, which requires knowing an optimal policy. It is intractable to find a globally optimal policy, so ERMAS instead finds a locally optimal policy by searching within a local neighborhood (trust region).
>
> *“Figure four does not appear to be referenced …”*
> Figure 4 is referenced in Section 4.3.
>
> *“… why you would want bounded regret.”* Could you clarify what you meant by “bounded regret”? We do not believe we use that term.

---

> > ### Comment · AnonReviewer5 · 2020-11-19
> > **Feedback**
> >
> > Thank you for the responses. I look forward to the revised version.
> >
> > The extended explanation of the bimatrix game is helpful. Has this been revised in the paper? It will help make the paper more accessible to the community if the paper is more self-contained.
> >
> > "method just using TRPO" I understand that this is not what is being done in the paper. I was suggesting it would help if this was made more clear in the paper.

---

> > > ### Author Response · Authors · 2020-11-25
> > > **Update on revisions**
> > >
> > > In response to your question: Yes, the sections involving the bimatrix game have been substantially revised, following your feedback. We feel that the clarity of these sections has improved significantly as a result.
> > >
> > > As you pointed out, it is important that the reader has a clear understanding of the motivation behind our technical contributions and the types of use-cases where they would naturally apply. We have added a new section to the Appendix discussing this, which we hope will complement the technical contributions developed in the main text. We have reproduced this section in our “General comments” reply, above. We intend to incorporate elements of this discussion into the Introduction when preparing a camera-ready version.
> > >
> > > Thanks again for your feedback and discussion.

---

> > > > ### Comment · AnonReviewer5 · 2020-11-25
> > > > **Re: Revisions**
> > > >
> > > > These revisions are appreciated. They have helped clarify the motivation, experiments and method. I agree with R1 that additional experiments would add greatly to the paper. I have raised my score on the paper.

---

### Author Response · Authors · 2020-11-18
**General Response to Reviewers' Comments**

We thank all reviewers for their thoughtful feedback. We have posted a revision to incorporate some of your feedback and will address the remaining concerns soon in another revision.

A high-level summary of the feedback:
* All reviewers agree that robustness in multi-agent reinforcement learning (MARL) is an important problem.
* The experiments are comprehensive and establish that ERMAS works well (R1, R3).
* The writing and structure are clear (R1, R3, R4), while R5 suggests additional exposition is needed, e.g., more clearly motivating and clarifying several design choices in ERMAS regarding uncertainty sets, our experimental designs, and interpretation of the results.
* R3 describes our empirical evaluation as ‘comprehensive’ while also suggesting having more baselines, and R1 suggests evaluating in another environment.
    * We already evaluate all the robustness baselines that are feasible for our experimental setting and applicable to the types of perturbations that we train agents to be robust to, i.e., changes in agent behavior due to unseen reward perturbations.
    * We are planning to run more experiments in a small multi-agent particle world environment. Given the computational requirements of MARL experiments, we hope to complete this before the end of the discussion phase and will share results here if feasible.
    * Please note that our MARL experiments also include two-level RL problems: a form of adaptive mechanism design where a planner agent influences the learning objective of the other agents. This introduces an additional level of difficulty for robust MARL algorithms that is novel in the robust reinforcement learning literature, and that only ERMAS can scale to.

We’d like to summarize our contributions and provide more intuition and context for our work.
* ERMAS is an MARL algorithm for training agent policies that are robust to uncertainty about the objectives of other agents. For instance, agents in a test environment might behave differently because they are more risk-averse than agents in the training environment. This is especially relevant for real-world AI agents that need to interact with humans, including self-driving cars interacting with human drivers, economic policymaking, and human-robot interaction.
* ERMAS is a form of robust optimization as it optimizes a robust policy while considering the worst-case impact on its objective. ERMAS finds this worst-case by training the other agents in the environment to act adversarially against the robust policy, while still behaving (approximately) optimal for their own objective. This behavior is encoded by approximate equilibria, akin to Nash equilibria. ERMAS is successful, even though there can be many or potentially infinite equilibria.

These considerations motivate the structure of ERMAS.
* ERMAS modifies the learning objective of the “non-robust” agents such that they are encouraged to find behaviors that minimize the robust agent’s reward while still being nearly optimal with respect to their original, selfish reward. This is formalized using epsilon-equilibria, where epsilon controls how much non-robust agents can sacrifice their own reward in order to decrease the robust agent’s reward.
* ERMAS bounds the adversariality of other agents by estimating how suboptimal their policies are. It estimates this suboptimality with a local approximation of regret: compare the selfish reward of the current agent policy to the most selfish policy within a local neighborhood of the policy space. To find this most selfish nearby policy, ERMAS uses trust-region optimization methods akin to TRPO and PPO.

---

> ### Author Response · Authors · 2020-11-24
> **Clarification of ERMAS motivation**
>
> We want to explain how ERMAS might be applicable in the real-world. To this end, we will revise our manuscript today to add a new Motivation Section to the Appendix (reproduced below) detailing additional high-level motivation and example use-cases for our method.
>
> **Motivation Section:**
> Many behavioral models, e.g., for self-driving cars or economic policymaking, are trained using RL in simulated environments, because real-world experiments are too expensive, infeasible, or unsafe. However, many RL policies need to interact with other agents whose real-world behavior might differ from that in the simulation. For example, a self-driving car trained in simulation needs to drive in traffic with human drivers.
>
> Often, the “reality gap” between simulated and real-world agents can be described as a difference in reward functions, e.g., humans might be more risk-averse than AI agents. However, these reality gaps can be hard to precisely quantify, as it is hard to learn the exact reward function of humans. Therefore, robust agents should be robust to uncertainty about the objectives of other agents.
>
> ERMAS is an adversarial robustness solution. ERMAS uses uncertainty sets that describe all “realistic” perturbations of agent reward functions (and hence their resulting behaviors). Consider a self-driving car in traffic. Suppose a Nash equilibrium is for all agents to drive at 70 mph, and a self-driving car has learned in simulation all rational agents would drive at 70 mph. A robust self-driving car needs to account for a situation where a more risk-averse human driver drives at 60 mph, whose ‘irrational’ behavior is optimal for a reward function that is different from that used in the simulation.
>
> ERMAS’s uncertainty set is bounded by the requirement that the simulation’s optimal policy be close to optimal even under a perturbed reward function. In our previous example, a “realistic” perturbation of the driver’s risk aversion is one where driving at 70mph is not preferable but also not unthinkable. Formally, this upper-bounds the statistical regret of driving at 70mph by some value ‘epsilon’, where the regret of a policy is defined by subtracting the policy’s reward from the optimal policy’s reward.
>
> The primary technical challenge of ERMAS is efficiently solving for the worst-case perturbation in this bounded uncertainty set. It does this by dualizing the robustness objective, yielding an optimization objective similar to constrained reinforcement learning. However, optimizing for this objective is difficult as it requires knowing whether a given agent policy is in the uncertainty set. Following the definitions, this requires us to estimate the statistical regret of agent policies which ERMAS avoids by instead estimating regret only within a local region of the policy space. In our previous example, ERMAS estimates the “realism” of a perturbation by only comparing driving at 70mph with the options of driving between 65-75mph, rather than all possible values from 0-100mph.

---

### Decision · Program_Chairs · 2021-01-07
**Final Decision**

**Decision:**

Reject

**Comment:**

The paper presents a multi-agent RL algorithm where the rewards of the other agents are only known up to some accuracy. The setting is somewhat restrictive, in the sense that the transition is assumed to be known. It would perhaps have been more interesting for the paper to also consider unknown transitions, so as to bring it closer to work in single-agent reinforcement learning. It also seems to not be making a very good job of linking the related work to the contribution of this paper (even after looking at the appendix).

- Authors briefly say in the introduction
"Alternative frameworks improve robustness, e.g., to changes in environment dynamics, observation or action spaces (Pinto et al., 2017; Li et al., 2019; Tessler et al., 2019), but do not address reality gaps due to reward function mismatches, as they use inappropriate metrics on the space of adversarial perturbations"

Authors should try and better explain the differences with those papers. Do  they consider changes in dynamics rather than the reward? It appears that the former is more general than the latter. Couldn't the authors compare with them with an appropriate experiment?

It is also hard to see how this exactly connects with a reality gap. What is the 'training' environment? What is the 'testing' environment? This is simply a robust optimisation algorithm applied to multi-agent games with partially unknown reward functions.

In addition the experiments themselves are not explained clearly.

On the plus side, I think the algorithmic details and experimental are interesting. If there was a better explanation and discussion/comparison with related work, then it would have been a good paper. Authors are encouraged to make a stronger effort to compare with other methods both in terms of the algorithm and experimentally.